# Maternal Fat-1 Transgene Protects Offspring from Excess Weight Gain, Oxidative Stress, and Reduced Fatty Acid Oxidation in Response to High-Fat Diet

**DOI:** 10.3390/nu12030767

**Published:** 2020-03-14

**Authors:** Kristen E. Boyle, Margaret J. Magill-Collins, Sean A. Newsom, Rachel C. Janssen, Jacob E. Friedman

**Affiliations:** 1Department of Pediatrics, Section of Nutrition, University of Colorado Anschutz Medical Campus, Aurora, CO 80045, USA; 2Department of Pediatrics, Section of Neonatology, University of Colorado Anschutz Medical Campus, Aurora, CO 80045, USA; mmagillcollins@salud.unm.edu (M.J.M.-C.); Sean.Newsom@oregonstate.edu (S.A.N.); Rachel-Janssen@ouhsc.edu (R.C.J.); Jed-Friedman@ouhsc.edu (J.E.F.); 3Department of Obstetrics and Gynecology, University of New Mexico School of Medicine, Albuquerque, NM 87106, USA; 4School of Biological and Population Health Sciences, Oregon State University, Corvallis, OR 97331, USA; 5Harold Hamm Diabetes Center, University of Oklahoma Health Sciences Center, Oklahoma City, OK 73104, USA

**Keywords:** omega-3, maternal obesity, metabolism, fatty acid oxidation

## Abstract

Overweight and obesity accompanies up to 70% of pregnancies and is a strong risk factor for offspring metabolic disease. Maternal obesity-associated inflammation and lipid profile are hypothesized as important contributors to excess offspring liver and skeletal muscle lipid deposition and oxidative stress. Here, we tested whether dams expressing the fat-1 transgene, which endogenously converts omega-6 (n-6) to omega-3 (n-3) polyunsaturated fatty acid, could protect wild-type (WT) offspring against high-fat diet induced weight gain, oxidative stress, and disrupted mitochondrial fatty acid oxidation. Despite similar body mass at weaning, offspring from fat-1 high-fat-fed dams gained less weight compared with offspring from WT high-fat-fed dams. In particular, WT males from fat-1 high-fat-fed dams were protected from post-weaning high-fat diet induced weight gain, reduced fatty acid oxidation, or excess oxidative stress compared with offspring of WT high-fat-fed dams. Adult offspring of WT high-fat-fed dams exhibited greater skeletal muscle triglycerides and reduced skeletal muscle antioxidant defense and redox balance compared with offspring of WT dams on control diet. Fat-1 offspring were protected from the reduced fatty acid oxidation and excess oxidative stress observed in offspring of WT high-fat-fed dams. These results indicate that a maternal fat-1 transgene has protective effects against offspring liver and skeletal muscle lipotoxicity resulting from a maternal high-fat diet, particularly in males. Altering maternal fatty acid composition, without changing maternal dietary composition or weight gain with high-fat feeding, may highlight important strategies for n-3-based prevention of developmental programming of obesity and its complications.

## 1. Introduction

The prevalence of obesity in U.S. women of reproductive age is over 35% and continues to grow [1]. Across multiple species, maternal obesity and high-fat feeding results in excess adiposity in fetal and/or adult offspring, both in adipose tissue depots and as ectopic fat stores in liver and skeletal muscle [2,3,4,5,6,7]. Such ectopic fat deposition is well-documented in adults with established obesity and is linked to perturbations in overall metabolic health, including inflammation, oxidative stress, and reduced mitochondrial metabolism. At the cellular level, animal models demonstrate that obesity during pregnancy accelerates fetal adipogenesis and alters lipid balance and energy metabolism, in multiple tissues [7,8]. Obesity is associated with a chronic state of low-grade inflammation. However, the role of maternal obesity-mediated alterations in inflammatory processes as a mechanism underpinning metabolic development in offspring is less understood. 

In adult animals and humans with established obesity, chronic low-grade inflammation is implicated in the metabolic consequences of obesity, including oxidative stress, insulin resistance, and dysregulated fuel metabolism [9,10]. In obese ewes, oxidative stress is hypothesized to contribute to fetal skeletal muscle adipose deposition and altered metabolic function [7,11], while in non-human primates, we have previously reported that maternal high-fat diet during pregnancy resulted in fetal hepatic lipid accumulation, oxidative stress, and skeletal muscle mitochondrial dysfunction in juvenile offspring [4,8]. Exposure to a maternal high-fat diet in utero results in a marked increase in the plasma omega-6 (n-6):omega-3 (n-3) polyunsaturated fatty acid (PUFA) ratio, accompanied by a pro-inflammatory cytokine profile [12]. Omega-6 fatty acids, such as linoleic acid or arachidonic acid, give rise to pro-inflammatory prostaglandins and leukotrienes when metabolized in vivo, while n-3 PUFA, such as α-linolenic acid or decosahexanaenoic acid, combat these with their anti-inflammatory function. Moreover, n-6 and n-3 fatty acids may directly affect fatty acid metabolism by targeting transcriptional regulators such as peroxisome proliferator-activated receptor (PPAR)-α [13], which may have lasting effects on adiposity and lipid metabolism across generations. In support of this, epidemiological studies in humans show that higher n-6 or lower n-3 PUFA levels during pregnancy and lactation are associated with greater offspring adiposity in childhood [14,15,16].

We previously reported that increasing the maternal and placental n-3:n-6 PUFA ratio mitigates many of the adverse programming effects of maternal high-fat diet on placenta inflammation and offspring liver and adipose tissue [17,18]. In these studies, we used the fat-1 (F1) transgenic mouse model, which endogenously converts n-6 PUFA to n-3 PUFA via ubiquitous expression of a novel *Caenorhabditis elegans*-derived n-3 desaturase under the β-actin promoter [19]. The fat-1 mouse has increased tissue levels of n-3 PUFA and n-3 PUFA-derived lipids [19,20]. Consistent with greater n-3 PUFA, fat-1 mice demonstrate reduced inflammation and less insulin resistance with high-fat feeding, despite the same dietary composition of n-3 and n-6 PUFAs [18,20,21,22]. Here, we hypothesized that a maternal high-fat diet would perturb oxidative metabolism and oxidative stress markers in fetal liver as well as adult skeletal muscle tissues, and that WT offspring of fat-1 transgenic dams would be protected from such deleterious effects of high-fat diet.

## 2. Materials and Methods

### 2.1. Ethics Statement

All animal studies were approved by the University of Colorado Institutional Animal Care and Use Committee and carried out in strict accordance with the guidelines set forth by the Guide for the Care and Use of Laboratory Animal by the National Institutes of Health.

### 2.2. Animal Breeding and Diet

Animals were treated as described [17]. Briefly, transgenic fat-1 C57BL6/J male mice were provided courtesy of Dr. J.X. Kang [19], and bred to wild type (WT) C57BL6/J females to obtain roughly 50% transgene-positive mixed litters. Eight-week-old female littermates, either WT or hemizygous fat-1, were then placed on a 45% kcal from fat diet (high-fat diet; Research Diets D12451) or a 10% kcal from fat control diet (Research Diets D12450B; WT only) and fed ad libitum for 8 wks prior to mating with a WT chow-fed male. Males were removed postcoitous, and mothers were maintained on respective diets throughout pregnancy and lactation. The fatty acid composition of the diets is listed in Appendix A.

### 2.3. Tissue Collection

For fetal collection, at day 18.5 of gestation, dams were fasted for 4 h in a clean cage. Dams were then euthanized by isoflurane anesthesia followed by exsanguination by cardiac puncture. Fetal-placental units were then quickly dissected as described [17]. Fetal livers were either processed immediately for fatty acid oxidation assays or flash frozen in liquid nitrogen for measures of enzyme activity. For analyses, each fetal group had representation from at least 4 dams.

For adult offspring measures, dams were allowed to deliver and pups were weaned onto either control or a high-fat diet to create six adult offspring groups based on three levels of maternal genotype/diet (maternal group) and 2 levels of post-weaning diet. Litters were standardized to 8 pups on day 1 to control for nutrient bias during lactation. Offspring genotype and sex were determined by tail-DNA PCR for presence or absence of *Fat-1* and *Sry* genes, respectively, as described [17]. Only WT offspring from fat-1 dams were analyzed and each offspring group had representation from at least 4 dams with male and female siblings stratified to either control or high-fat diet post-weaning. At 20 wks of age, animals were fasted for 5 h, then euthanized by isoflurane anesthesia followed by exsanguination by cardiac puncture. Liver and mixed gastrocnemius skeletal muscle samples were dissected free from visible adipose and connective tissues and either processed immediately for fatty acid oxidation assays or flash frozen in liquid nitrogen for subsequent analyses. Results from dams and select offspring measures have been reported previously. The study design is shown in Figure 1.

### 2.4. Body Weight and Insulin Tolerance Test

Offspring weight was recorded at weaning and every other week thereafter. At 20 weeks of age, animals were fasted for 5 h, fasting blood glucose was obtained from the tail vein and recorded by glucometer, and an insulin tolerance test was performed, as described [17]. Briefly, a bolus of recombinant insulin was injected i.p. at 0.0075 U/kg body weight and blood glucose readings were taken from the tail at 0, 10, 20, 30, 45, 60, 75, and 90 min post-injection. The glucose ‘fall from baseline’ over the first 30 min of the insulin tolerance test was used as an index of insulin sensitivity and was calculated as the area over the curve from fasting to the 30-minute glucose measure. 

### 2.5. Skeletal Muscle Triglyceride Quantitation

Muscle triglyceride content was measured by the modified Bligh and Dyer method [23]. Approximately 60 mg skeletal muscle tissue was homogenized in 1 mL of ice-cold methanol and lipids were extracted using 1:2 methanol:chloroform. The polar and non-polar phases were then separated by centrifugation, and the lower non-polar phase was isolated and dried under nitrogen. The dried samples were then resuspended in isopropanol + 2% Triton X-100 for direct triglyceride quantitation using Infinity Triglycerides Reagent (Thermo Scientific, Waltham, MA, USA) and a standard curve with glycerol standard solution (Sigma-Aldrich, St. Louis, MO, USA) per manufacturer’s instructions. Resultant triglyceride concentrations were normalized to starting tissue weight. Results from dams and select offspring measures have been reported previously, as were fetal and adult liver triglyceride content [17].

### 2.6. Fatty Acid Oxidation

Fatty acid oxidation was measured in fresh tissue samples as previously described [24]. Approximately 30 mg of fresh skeletal muscle tissue, or one fetal liver were collected in sucrose-EDTA buffer. Samples were minced thoroughly with scissors and then diluted 20-fold with additional sucrose-EDTA buffer. Tissue was homogenized on ice using Teflon on glass, then added to the incubation wells of a modified 48-well plate with a channel cut between the adjacent trap wells, and then sealed airtight. Incubation buffer containing [1-^14^C]-palmitate (PerkinElmer Life Sciences, Waltham, MA, USA) was added to initiate the reaction. Following 45 min of incubation at 37 °C, 70% perchloric acid was added to terminate the reaction and liberate the accumulated ^14^CO_2_ to the sodium hydroxide trap wells, which were sampled for label incorporation into ^14^CO_2_ (complete fatty acid oxidation). Acidified tissue samples were sampled for ^14^C-acid soluble metabolites (ASMs). Complete fatty acid oxidation and ASMs were summed for total fatty acid oxidation. Tissue homogenates were assayed for total protein content by bicinchoninic acid assay for data normalization. All measures were performed in quintuplet.

### 2.7. Enzyme Activity Assays and Glutathione Redox State

Mitochondrial-enriched supernatants were prepared from frozen fetal liver or skeletal muscle samples and enzyme activity of citrate synthase and aconitase were measured spectrophotometrically, as described [25]. The protein content of each sample was determined by bicinchoninic acid assay. All enzyme activities were normalized to the total protein content. Overall oxidative stress was measured by total and oxidized glutathione (GSH and GSSG, respectively) from deproteinated skeletal muscle samples using a commercially available kit (Caymen Chemical, Ann Arbor, MI, USA).

### 2.8. Protein Measures

Total protein content of skeletal muscle was determined by BCA assay. Specific protein content of manganese superoxide dismutase (MnSOD) and sirtuin 3 (Sirt3) were analyzed by Western blot, with glyceraldehyde 3-phosphate dehydrogenase (GAPDH) as reference control, as described [26]. All results were expressed relative to the loading control and to the mean for WT-CD animals. Specific antibodies used are listed in Appendix A.

### 2.9. Data Analysis

For offspring measures involving a time course (weight gain, insulin tolerance test), 2-way repeated measures one-way analysis of variance (ANOVA) was used, with Bonferroni *post hoc* analyses to compare replicate means. These time course analyses were performed in Graphpad Prism. All other analyses were performed in R. For single time point measures, each variable was assessed for homogeneity of variance using Levene’s test with Brown–Forsythe modification. Where appropriate, log or Box-Cox transformed data were used for statistical analyses with normality verified using the Shapiro–Wilk test of the residuals. To assess fetal tissue measures across maternal group, ANOVA was run with Tukey post-hoc analyses denoting significant differences between groups. To assess the six offspring groups in adult tissues, we first tested for main effects of post-weaning diet or maternal group, or an interaction of the two using 2 × 3 ANOVA with two levels of post-weaning diet and three levels of maternal group. Significant interactions are reported with Tukey post-hoc analyses denoting significant differences between groups. Where interactions were not significant, the main effects were interpreted. For a significant effect of the maternal group, Tukey post-hoc analyses denote significant differences between groups consolidated across post-weaning diet. Significant effects of post-weaning diet were reported. All data are expressed as the mean ± standard deviation (SD) and statistical significance was indicated at *p* ≤ 0.05. 

## 3. Results

### 3.1. Maternal Fat-1 Transgene Improves Oxidative Stress Markers in Fetal Liver

We previously reported that fetal liver from fat-1 high fat-fed (FAT-HF) dams had elevated the phospholipid n-3/n-6 PUFA ratio compared with WT-CD and WT-HF, and that the fetal livers from WT-HF dams showed elevated triglyceride content, which was rescued in the FAT-HF group [17]. To determine whether the maternal fat-1 transgene protects the fetal liver from the deleterious effects of maternal high-fat diet, we measured aconitase enzyme activity as an index of oxidative stress and observed elevated activity in liver from FAT-HF fetuses (Figure 2a). Aconitase is sensitive to redox damage, thus higher activity in FAT-HF fetuses is indicative of reduced oxidative stress. To evaluate differences in lipid metabolism, we measured mitochondrial fatty acid oxidation in fresh fetal livers from the same dams. We observed a similar trend for greater fatty acid oxidation in the WT-HF group. Total fatty acid oxidation (Figure 2b) reflects the sum of complete oxidation of fatty acids to CO_2_ (Figure 2c) and partially oxidized fatty acids (Figure 2d) that have been solubilized in acid. These measures both showed similar patterns of lower measures in offspring of WT-HF dams compared with offspring of either WT-CD or FAT-HF dams, though these did not reach statistical significance (ANOVA *p* ≥ 0.056). 

### 3.2. Maternal Fat-1 Transgene Partially Protects Offspring from Excess Weight Gain and Insulin Resistance

To determine whether the maternal fat-1 transgene protects offspring from the deleterious effects of maternal high-fat diet, we compared sibling offspring from WT-CD, WT-HF, and FAT-HF dams. Male and female sibling pairs were weaned on to either control or high-fat, to determine the effect of the maternal fat-1 transgene on the offspring response to post-weaning diet. Despite similar body mass at weaning, there was a significant effect of maternal high-fat diet on increased weight by 20 weeks of age for both male and female offspring, regardless of maternal genotype (Figure 3c,f). A significant effect of maternal group (WT-CD, WT-HF, FAT-HF) on weight gain in male and female offspring, regardless of post-weaning diet, was also observed (Figure 3). Males from FAT-HF mothers in particular were protected from post-weaning high-fat diet-induced weight gain at 8 wks through 18 wks of age (Figure 3b). Similar, observations were made in female offspring of FAT-HF dams, where offspring were partially protected on post-weaning high-fat diet relative to offspring of WT-HF dams, though not as robustly as their male counterparts (Figure 3d–f).

Insulin tolerance tests were performed with bolus i.p. insulin injection at 20 wks of age in the male and female WT offspring of WT-CD, WT-HF, and FAT-HF dams. Male offspring had similar fasting blood glucose levels (0 min), regardless of maternal group (Figure 4a,b). Female WT-HF offspring had elevated fasting glucose levels relative to WT-CD and FAT-HF offspring on a post-weaning control diet (Figure 4d) and elevated glucose levels relative to WT-CD offspring on a post-weaning high-fat diet (Figure 4e). Female offspring showed a significant effect of maternal group on glucose levels over time, where elevated blood glucose was observed in offspring of WT-HF dams, while offspring of FAT-HF dams tended to track with offspring of WT-CD dams (Figure 4d,e). Similar observations were made for male offspring on a post-weaning control diet, but no maternal group differences were observed for male offspring on a post-weaning high-fat diet (Figure 4a,b). To test for the effects of maternal group, post-weaning diet, and the interaction in all animals, we calculated the insulin tolerance test fall from baseline over 30 min as a single measure representing insulin effects on glucose disposal. In male offspring, there was a significant effect of maternal group, where offspring of WT-HF had less robust response to insulin bolus than WT-CD offspring (Figure 4c). This was not observed in the female offspring (Figure 4f). Males also appeared to have more profound effects of post-weaning diet on fall from baseline, where offspring on a post-weaning high-fat diet demonstrated lower fall from baseline compared with those on a post-weaning control diet (Figure 4c).

### 3.3. Maternal Fat-1 Transgene Protects Offspring from High-Fat Diet Reduced Liver Fatty Acid Oxidation

We have previously shown that liver triglyceride content was elevated in adult male offspring of WT-HF dams, and this effect of maternal high-fat diet was ameliorated in offspring of FAT-HF dams [17]. To determine whether maternal high-fat diet alters fatty acid oxidation in adult male and female offspring challenged with a high-fat diet, we measured fatty acid oxidation in the liver of WT offspring weaned to either control or high-fat diet. We observed that adult liver fatty acid oxidation was reduced in offspring of WT-HF dams, while offspring of FAT-HF dams showed partial protection from this effect (Figure 5a). We observed no effect of offspring post-weaning diet on these measures. Individually, liver oxidation of fatty acids completely to CO_2_ (Figure 5b) was not different between groups, though partially oxidized lipids showed similar patterns of lower measures in offspring of WT-HF dams compared with offspring of either WT-CD or FAT-HF dams (Figure 5c).

### 3.4. Maternal Fat-1 Transgene Protects Offspring from Elevated Skeletal Muscle Triglyceride and Reduced Fatty Acid Oxidation

We next measured skeletal muscle triglyceride content and observed elevated levels in the offspring of WT-HF dams, compared with those of WT-CD dams. This effect was improved in offspring of FAT-HF dams (Figure 6a), regardless of whether offspring were on a control or high-fat diet. There was a significant effect of offspring post-weaning diet, driven by offspring of WT-HF dams, where skeletal muscle triglycerides were further elevated in the offspring on post-weaning high-fat diet. Little to no effect of post-weaning diet was observed for the WT-CD or FAT-HF offspring (Figure 6a). 

We next investigated skeletal muscle fatty acid oxidation and, similar to liver, we observed reduced fatty acid oxidation in offspring of WT-HF, but not FAT-HF dams (Figure 6b). Both complete oxidation of fatty acids to CO_2_ and incomplete oxidation showed similar patterns of lower measures in offspring of WT-HF dams compared with offspring of either WT-CD or FAT-HF dams (Figure 6c,d). To test whether differences in mitochondrial content played a role in these metabolic patterns, we measured citrate synthase enzyme activity in the skeletal muscle tissue but did not observe any differences (Appendix A).

### 3.5. Maternal Fat-1 Transgene Partially Protects from Perturbations in Antioxidant Defense and Redox Balance

We measured aconitase enzyme activity in adult offspring skeletal muscle tissue as an index of oxidative stress, though there were no differences between groups (Figure 7a). To determine overall cellular redox, we measured the reduced and oxidized glutathione ratio (GSH:GSSG), where a lower redox balance suggests greater oxidative stress. We found offspring of WT-HF dams had a lower redox balance than offspring of WT-CD dams (Figure 7b). Though there were not significant differences between FAT-HF offspring and either WT-CD or WT-HF offspring, FAT-HF values tracked more closely with WT-HF offspring values (Figure 7b). To determine differences in mitochondrial antioxidant defense in offspring skeletal muscle, we measured protein content of sirtuin (SIRT) 3 and manganese superoxide dismutase (MnSOD), a mitochondria-localized antioxidant enzyme. SIRT3 protein content was elevated in WT-HF offspring relative to WT-CD offspring, and partially rescued in offspring of FAT-HF dams (Figure 7c). MnSOD was also elevated in offspring of WT-HF dams compared with offspring of WT-CD and FAT-HF dams, an effect that was exacerbated when WT-HF offspring were fed a post-weaning high-fat diet (Figure 7d).

## 4. Discussion

We previously reported that increasing the maternal n-3/n-6 fatty acid ratio, which protects from excess inflammation in dams and placenta from fat-1 animals, partially or fully protected offspring of high-fat-fed dams from excess liver triglycerides and adiposity, and adipose tissue inflammation in adulthood [17]. Here, we show that offspring of fat-1 dams exhibit evidence of reduced oxidative stress in fetal liver and are protected from excess skeletal muscle triglyceride accumulation in adulthood, regardless of whether they were fed a post-weaning control or high-fat diet. In addition, offspring of FAT-HF dams were protected from reduced skeletal muscle and liver fatty acid oxidation or excess oxidative stress observed in offspring of WT-HF dams. Thus, by altering maternal fatty acid composition, without any change in maternal weight gain, liver and skeletal muscle metabolic adaptations were mitigated in adult WT offspring of fat-1 dams.

There is no consensus on a single mechanism responsible for the developmental outcomes associated with maternal obesity. However, fat-1 dams gained similar amounts of weight on the high-fat diet but were protected from maternal and placental inflammation, suggesting that other maternal factors not protected by the fat-1 transgene, such as hyperinsulinemia [17] or potential stressors prior to placental implantation contribute to the observed offspring outcomes. Here, we show that the WT offspring from fat-1 dams partially protects male offspring from excessive weight gain on a post-weaning high-fat diet to 18 wks of age, though these differences were less robust at 20 wks. Though all male offspring had similar body weight at weaning (4 wks), by 14 wks of age there were notable differences between male offspring of WT-CD, WT-HF, and FAT-HD dams weaned to high-fat diet. This protection from excess weight gain was not observed in male offspring weaned to control diet, nor in female offspring, where FAT-HF offspring tended to track with WT-HD offspring. Nevertheless, in both male and female offspring weaned to control diet, the maternal fat-1 transgene conferred some protection from insulin resistance, where WT-HF offspring displayed elevated glucose levels during the insulin tolerance test. This effect was more prominent in offspring weaned to control diet. On the whole, the glucose fall from baseline during first 30 minutes of the insulin tolerance test showed that male offspring of fat-1 dams had protection from insulin resistance, whereas female offspring did not.

Fetuses have limited capacity for fatty acid oxidation and lipid fuel overload may result in excess fat storage [27,28]. Ectopic fat storage in the liver and skeletal muscle is a classic characteristic of obesity and insulin resistance and has been observed in the liver of human babies born to mothers with obesity and gestational diabetes [3]. In larger mammals (e.g. sheep, primates), skeletal muscle and liver from fetuses of obese dams are also characterized by disruption to insulin signaling, excess oxidative stress and inflammation, and deficits in oxidative metabolism and nutrient sensing pathways when compared with fetuses of control-fed dams [4,7,8,11]. Japanese macaques on a high-fat diet also exhibit oxidative damage in fetal liver during the early 3rd trimester [4]. Consistent with previous studies, our mouse model of obese pregnancy induced greater weight gain and ectopic lipid accumulation in the fetal liver of the offspring [17]. Aconitase, a citric acid cycle enzyme, also acts as a superoxide sink, similar to MnSOD. This neutralizes superoxide, but also inactivates the enzyme [29]. Thus, reduced aconitase enzyme activity is a good index of oxidative stress. Although we did not observe greater oxidative stress in WT-HF versus WT-CD fetal livers, as determined by aconitase activity, we did find fetal liver from FAT-HF dams had evidence of lower oxidative stress indicating some protection in the offspring of fat-1 dams.

We previously reported that the liver of adult offspring of WT-HF-fed dams have greater triglyceride content compared with WT-CD, which was ameliorated in the offspring of FAT-HF dams [17]. Here, we show that liver lipid oxidation is reduced in the adult offspring of WT-HF-fed dams, regardless of post-weaning diet. This was partly normalized in the offspring of FAT-HF dams, which is in line with our observations in skeletal muscle tissue of the adult offspring, where triglycerides are elevated and fatty acid oxidation is reduced in WT-HF offspring and ameliorated in offspring of FAT-HD dams. In general, these effects are independent of post-weaning diet, though elevated skeletal muscle triglyceride content was exacerbated in WT-HF offspring on a post-weaning high-fat diet. Similar observations of elevated lipid storage and transcriptional or protein evidence of reduced oxidative metabolism have been observed in skeletal muscle and liver tissue from fetal and adult offspring high-fat-fed dams [30,31]. Fewer groups have directly measured oxidative metabolism in liver or skeletal muscle from offspring of high-fat-fed dams. Mouse models have shown elevated lipid stores and/or reduced mitochondrial respiration of non-lipid substrate in gastrocnemius muscle from offspring of high-fat-fed dams [2,32]. Our findings in a well-established non-human primate model suggest that maternal obesity or Western style diet, alone or in combination, leads to decreased oxidative metabolism in offspring muscle [8]. These fetuses also had reduced mitochondrial content, diminished oxidative capacity, and lower mitochondrial efficiency in muscle. Importantly, switching obese mothers to a healthy diet prior to pregnancy did not improve fetal muscle mitochondrial function, suggesting that obese maternal environment, characterized by hyperlipidemia and an exaggerated state of inflammation and oxidative stress [8], are important drivers of skeletal muscle dysfunction in the offspring.

## 5. Conclusions

In conclusion, we confirm the strong association of n-3 fatty acids as protective factors in insulin resistance and weight gain in the offspring of high-fat-fed dams, through changes in oxidative metabolism. In addition, we found that exposure to greater maternal n-3/n-6 ratio during pregnancy and lactation influences oxidative stress in the fetus and in adult offspring liver and skeletal muscle. Some of these effects appear to have sex-specific influences in the developing offspring. The results of this study indicate that starting in early gestation, the maternal metabolite environment may have an important effect on offspring weight and potentially, long-term cardiometabolic risk, consistent with the developmental origins of health and disease hypothesis. Recent studies also suggest that n-3 PUFAs are associated with anti-obesity properties in part through the microbiome [33,34]. Others indicate there may be an attenuated systemic response to n-3 PUFA supplementation in pregnant women with overweight or obese [35]. While we used a transgenic approach to modify n-3 PUFA, understanding factors in obesity that may modulate response to n-3 supplementation deserves further attention in human studies.

## Figures and Tables

**Figure 1 nutrients-12-00767-f001:**
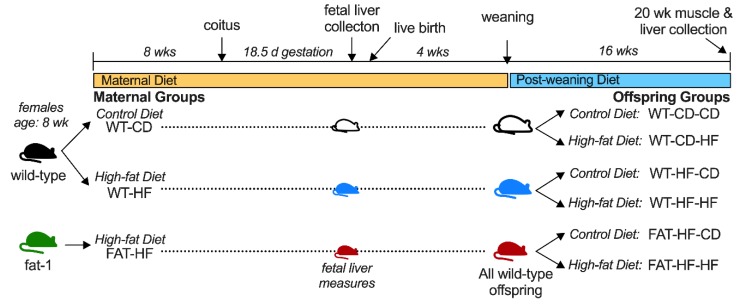
Study Design.

**Figure 2 nutrients-12-00767-f002:**
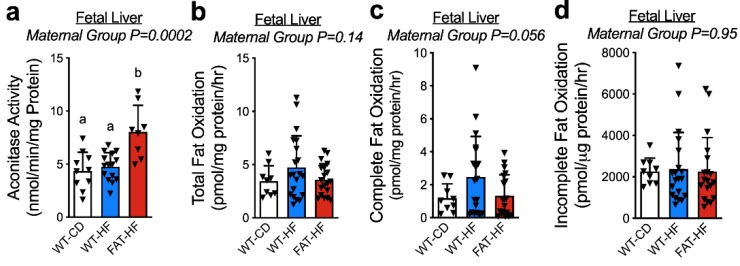
Maternal fat-1 transgene alters aconitase activity and complete fatty acid oxidation in fetal liver. Aconitase activity (**a**) and total fatty acid oxidation (**b**) were measured in fetal liver samples from WT dams fed a control diet (WT-CD), or offspring of WT or fat-1 hemizygous dams fed a high-fat diet (WT-HF and FAT-HF, respectively) throughout pregnancy, with sample collection at day E18.5. Total fatty acid oxidation is the sum of complete fatty acid oxidation to CO_2_ (**c**) and partially oxidized fatty acids (**d**). All data are presented as mean ± SD. Different letters above bars denote statistically significant differences between those bars at *p* < 0.05 significance level.

**Figure 3 nutrients-12-00767-f003:**
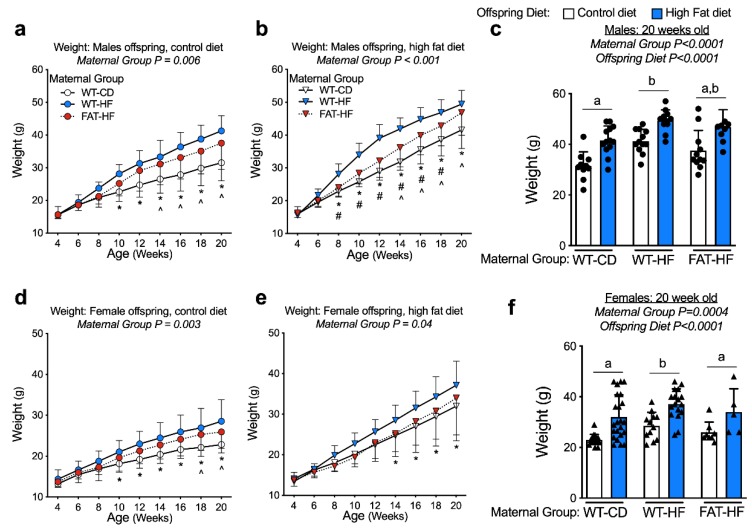
Maternal fat-1 transgene partially protects offspring from excess weight gain. Weight was measured from weaning to 20 wks in offspring of WT dams fed a control diet (WT-CD), or offspring of WT or fat-1 hemizygous dams fed a high-fat diet (WT-HF and FAT-HF, respectively) throughout pregnancy and lactation. All offspring were WT and were weaned to control or high-fat diet for male (**a**–**c**) or female (**d**–**f**) offspring. For males, n ≥ 10 per group. For females, n ≥ 5 per group. All data are presented as mean ± SD. * *p* < 0.05 for difference between WT-CD and WT-HF. ^ *p* < 0.05 for difference between WT-CD and FAT-HF. # *p* < 0.05 for difference between WT-HF and FAT-HF. Different letters above bars denote statistically significant differences between those bars at *p* < 0.05 significance level.

**Figure 4 nutrients-12-00767-f004:**
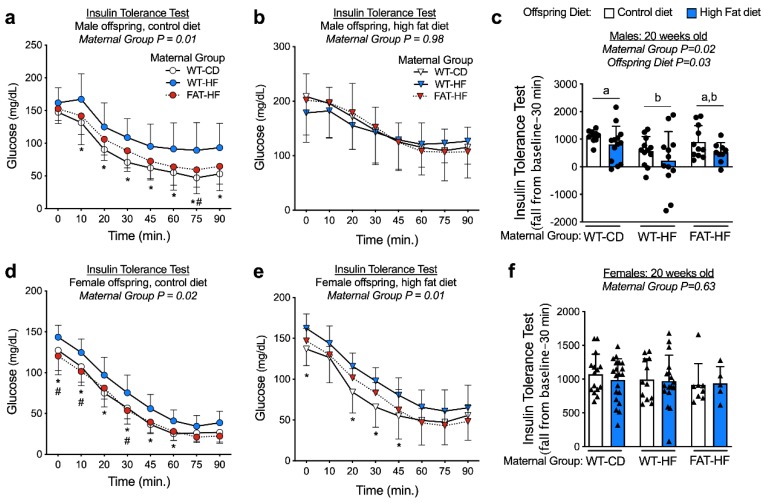
Maternal fat-1 transgene partially protects offspring from insulin resistance. Blood glucose levels were measured following bolus insulin injection (insulin tolerance test) in offspring of WT dams fed a control diet (WT-CD), or offspring of WT or fat-1 hemizygous dams fed a high-fat diet (WT-HF and FAT-HF, respectively) throughout pregnancy and lactation. All offspring were WT and were weaned to control or high-fat diet for male (**a**–**c**) or female (**d**–**f**) offspring. For **c, f**, data are the area fall from baseline over 30 minutes with baseline represented as fasting glucose levels. For males, n ≥ 9 per group. For females, n ≥ 5 per group. All data are presented as mean ± SD. * *p* < 0.05 for difference between WT-CD and WT-HF. Different letters above bars denote statistically significant differences between those bars at *p* < 0.05 significance level.

**Figure 5 nutrients-12-00767-f005:**
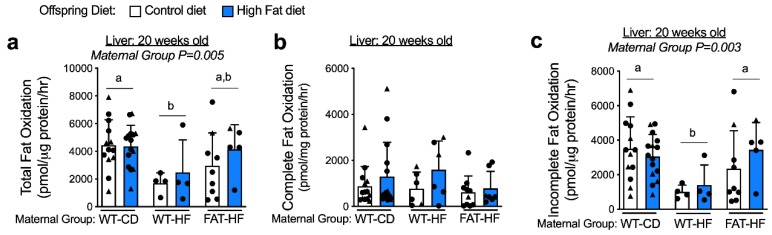
Maternal fat-1 transgene improves offspring liver fatty acid oxidation rates. Liver total fatty acid oxidation (**b**), complete fatty acid oxidation (**b**) and incomplete fatty acid oxidation (**c**) were measured at 20 wks in offspring of WT dams fed a control diet (WT-CD), or offspring of WT or fat-1 hemizygous dams fed a high-fat diet (WT-HF and FAT-HF, respectively) throughout pregnancy and lactation. All offspring were WT males (circles) and WT females (triangles) and were weaned to control or high-fat diet (post-weaning diet). Data are presented as mean ± SD. Different letters above bars denote statistically significant differences between those bars at *p* < 0.05 significance level.

**Figure 6 nutrients-12-00767-f006:**
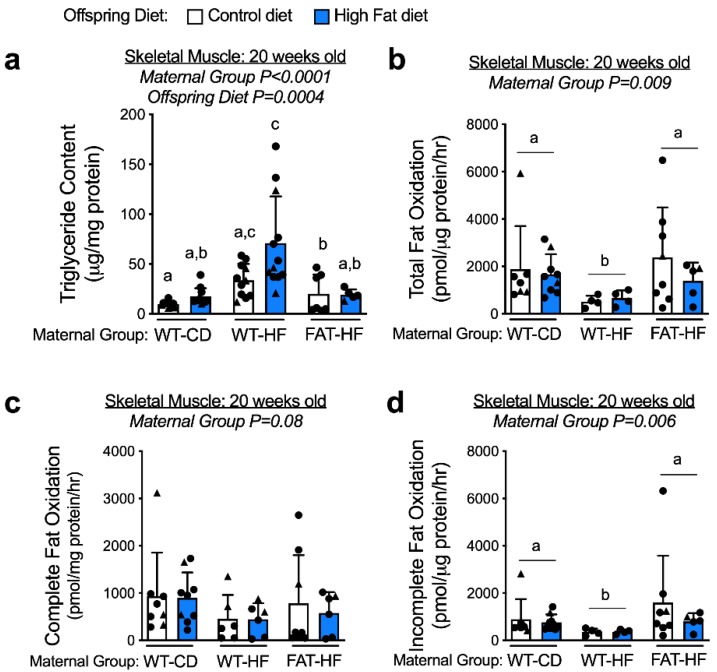
Maternal fat-1 transgene protects offspring from excess intramuscular triglyceride and lower fatty acid oxidation. Skeletal muscle triglyceride content (**a**), and total (**b**), complete (**c**), and incomplete (**d**) fatty acid oxidation were measured in offspring of WT dams fed a control diet (WT-CD), or offspring of WT or fat-1 hemizygous dams fed a high-fat diet (WT-HF and FAT-HF, respectively) throughout pregnancy and lactation. All offspring were WT males (circles) and WT females (triangles) and were weaned to control or high-fat diet (post-weaning diet). Data are presented as mean ± SD. Different letters above bars denote statistically significant differences between those bars at *p* < 0.05 significance level.

**Figure 7 nutrients-12-00767-f007:**
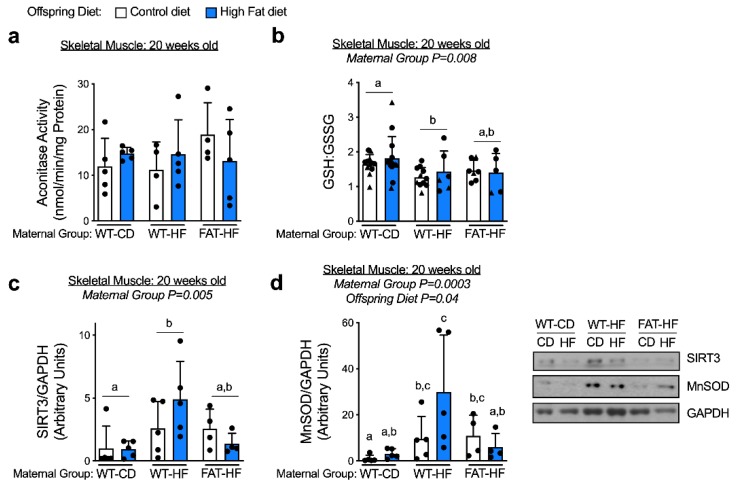
Maternal fat-1 transgene partially protects from perturbations in antioxidant defense and redox balance. Skeletal muscle aconitase enzyme activity (**a**), oxidized:reduced glutathione ratio (**b**), and SIRT3 (**c**) and MnSOD (**d**) protein content were measured in offspring of WT dams fed a control diet (WT-CD), or offspring of WT or fat-1 hemizygous dams fed a high-fat diet (WT-HF and FAT-HF, respectively) throughout pregnancy and lactation. All offspring were WT males (circles) and WT females (triangles) and were weaned to control or high-fat diet (post-weaning diet). Data are presented as mean ± SD. Different letters above bars denote statistically significant differences between those bars at *p* < 0.05 significance level.

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
