# Peer review of "Maternal Fat-1 Transgene Protects Offspring from Excess Weight Gain, Oxidative Stress, and Reduced Fatty Acid Oxidation in Response to High-Fat Diet"

_nutrients, 2020, doi:10.3390/nu12030767_

Round 1
Reviewer 1 Report
The Authors made an ambitious and interesting work to demonstrate how "an altering maternal fatty acid composition, without changing maternal dietary composition or weight gain with high fat feeding, may prevent the developmental programming of obesity and its complications".
This work has well conducted and results are very suggestive, however, there are some obvious amendments that should be done and are detailed here.
1) Authors have to provide a flowchart to better describe the various steps of the work, the experiments done at each step and the different categories studied.
2) The large quantity of abbreviations complicates both reading and comprehension of the text. It has to be reduced
3) The statistical analysis is well conducted in general, but Authors have to explain the following points
- a) “ 2-way repeated measures one-way analysis of variance (ANOVA)”. Is it 2-way or one-way?
- b) initially the Bonferroni post hoc analyses has been used but later have been substituted with the Tukey post-hoc analyses. Why?
- c) data are expressed as mean ± the standard error of mean (which indicates the precision of the mean). It is not correct. The have to be expressed as mean ±standard deviation (which indicates the data dispersion). It should be better to use the box plot graphs.
- d) Authors have not indicated the software used for the statistical analysis.
4) Figures are not well understandable.
- a) to many abbreviations
- b) the indication of statistically significant differences between groups is not clear and not well presented.
Author Response
The Authors made an ambitious and interesting work to demonstrate how "an altering maternal fatty acid composition, without changing maternal dietary composition or weight gain with high fat feeding, may prevent the developmental programming of obesity and its complications".
This work has well conducted and results are very suggestive, however, there are some obvious amendments that should be done and are detailed here.
We thank you for your helpful review. Specific comments below.
1) Authors have to provide a flowchart to better describe the various steps of the work, the experiments done at each step and the different categories studied.
This is now included as Figure 1 and all subsequent figures have been renumbered
2) The large quantity of abbreviations complicates both reading and comprehension of the text. It has to be reduced
We have reduced the abbreviations for fatty acid oxidation, insulin tolerance test, fall from baseline, post-weaning diet, maternal group, bicinchoninic acid assay, high-fat diet and control diet (except when referring to animal groups). We retained group identifier abbreviations for brevity and ease of reading, though if necessary we can change these too. Some other common abbreviations remain, but this has greatly improved the ease of reading.
3) The statistical analysis is well conducted in general, but Authors have to explain the following points
- a) “ 2-way repeated measures one-way analysis of variance (ANOVA)”. Is it 2-way or one-way?
We apologize, this was a typo from a previous draft. It is a repeated measures ANOVA
- b) initially the Bonferroni post hoc analyses has been used but later have been substituted with the Tukey post-hoc analyses. Why?
This was an oversight, we have now used Tukey post-hoc tests for all analyses with minimal/no change to results.
- c) data are expressed as mean ± the standard error of mean (which indicates the precision of the mean). It is not correct. The have to be expressed as mean ±standard deviation (which indicates the data dispersion). It should be better to use the box plot graphs.
We agree that SEM is not ideal, and this was why we included individual datapoints to show data dispersion for most graphs (Figs 1, 4, 5, 6). We chose not to include box plots because it would be difficult to see the details of the box plot when individual data points are included. We maintain that it is important to include individual datapoints to indicate male/female animals (circles/triangles). This information would be lost with summary box plots. We have now included datapoints for all bar graphs to demonstrate dispersion, and have now represented the data as mean +/- SD.
- d) Authors have not indicated the software used for the statistical analysis.
This information has been added to the analysis section (line 198-199)
4) Figures are not well understandable.
- a) to many abbreviations
We have removed most abbreviations and the only remaining are for maternal group names
- b) the indication of statistically significant differences between groups is not clear and not well presented.
We assume this is referring to the letters denoting groups different from each other. We agree this is cumbersome. The concept here is that columns with that do not share a letter marking are significantly different from each other. With up to 15 post-hoc tests in some cases, there is not really an easy way to present the data. This is a fairly common way of presenting significance in datasets such as these (examples shown below). We have attempted to clarify this in the figure legends and have simplified some graphs where only there are significant main effects, but no interactions. However, for data with significant interactions we kept them as is. We are open to other suggestions if further change is necessary.
Please see attached document for specific figures:
From: Chengzhe et al. Scientific Reports, 7: 5033, 2017: DOI:10.1038/s41598-017-04721-6 PMID: 28694497
From: Jayasundara and Somero. Journal of Experimental Biology 2013. 216: 2111-2121; doi: 10.1242/jeb.083873 PMID: 23678101

Reviewer 2 Report
The authors’ work is simply fantastic as the researcher of fatty acid metabolism also using fat-1 mice. I have two comments for the current version of the manuscript.
- Please provide the nutrient detail (composition table) that you used. Especially, you should write the composition for omega-3 and omega-6 fatty acids used in this research.
- I cannot simply understand the meaning of the characters (a, b, ab etc) used in your figures. Please explain the meaning clearly.
Author Response
The authors’ work is simply fantastic as the researcher of fatty acid metabolism also using fat-1 mice. I have two comments for the current version of the manuscript.
We thank you for your helpful review. Specific comments below.
- Please provide the nutrient detail (composition table) that you used. Especially, you should write the composition for omega-3 and omega-6 fatty acids used in this research.
We have now included this information in Supplementary Table 1.
- I cannot simply understand the meaning of the characters (a, b, ab etc) used in your figures. Please explain the meaning clearly.
As for response to comment 4.2.b from reviewer 1:
We agree this is cumbersome. The concept here is that columns with that do not share a letter marking are significantly different from each other. With up to 15 post-hoc tests in some cases, there is not really an easy way to present the data. This is a fairly common way of presenting significance in datasets such as these (examples shown below). We have attempted to clarify this in the figure legends and have simplified some graphs where only there are significant main effects, but no interactions. However, for data with significant interactions we kept them as is. We are open to other suggestions if further change is necessary.
Please see attached document for example figures.
From: Chengzhe et al. Scientific Reports, 7: 5033, 2017: DOI:10.1038/s41598-017-04721-6 PMID: 28694497
From: Jayasundara and Somero. Journal of Experimental Biology 2013. 216: 2111-2121; doi: 10.1242/jeb.083873 PMID: 23678101

Round 2
Reviewer 1 Report
The manuscript has undergone significant improvement and is worth to be published.